

# Dynamic sampling of images from various categories for classification based incremental deep learning in fog computing

Swaraj Dube, Yee Wan Wong and Hermawan Nugroho

Department of Electrical and Electronic Engineering, University of Nottingham - Malaysia Campus, Semenyih, Selangor, Malaysia

Corresponding author
Swaraj Dube,
kecy3dsm@nottingham.edu.my

## ABSTRACT

Incremental learning evolves deep neural network knowledge over time by learning continuously from new data instead of training a model just once with all data present before the training starts. However, in incremental learning, new samples are always streaming in whereby the model to be trained needs to continuously adapt to new samples. Images are considered to be high dimensional data and thus training deep neural networks on such data is very time-consuming. Fog computing is a paradigm that uses fog devices to carry out computation near data sources to reduce the computational load on the server. Fog computing allows democracy in deep learning by enabling intelligence at the fog devices, however, one of the main challenges is the high communication costs between fog devices and the centralized servers especially in incremental learning where data samples are continuously arriving and need to be transmitted to the server for training. While working with Convolutional Neural Networks (CNN), we demonstrate a novel data sampling algorithm that discards certain training images per class before training even starts which reduces the transmission cost from the fog device to the server and the model training time while maintaining model learning performance both for static and incremental learning. Results show that our proposed method can effectively perform data sampling regardless of the model architecture, dataset, and learning settings.

## INTRODUCTION

A lot of success in traditional deep learning has been attributed to access to large amounts of data and high computing power (*Lecun, Bengio & Hinton, 2015*). Even if large amounts of data are available, training deep learning models is challenging because powerful hardware resources are needed to do so, and training such deep models on large amounts of data is a highly time-consuming process because most of the deep learning training uses stochastic gradient descent and backpropagation for learning which is highly computationally expensive (*Sze et al., 2017*). This is why there is a clear need to develop algorithms that

can select only useful sets of data from the training dataset and still maintain the learning performance of models while reducing the training time.

In deep learning, data sampling can take place at two stages: before training and during training. In both cases, there are various parameters based on which data sampling takes place. E.g., when performing data sampling during training, samples are discarded based on whether or not certain samples are helping to improve the model learning process. In deep learning, it is often observed that the majority of the learning takes place during the first few epochs of training (*Katharopoulos & Fleuret, 2018*) which is why it makes sense to sample data after this stage that do not contribute to the overall learning of models. For performing data sampling before training, selected samples are usually chosen that are able to represent the overall data distributions of all samples before sampling. Model training can be greatly accelerated by performing data sampling during training; however, for further acceleration, data sampling must also take place before training begins. It is highly challenging to perform data sampling before model training because since deep neural networks are stochastic in nature, it is not possible to predict the learning trajectory of a deep learning model on a given dataset before training even starts. Once training starts after data sampling has been performed, the training process cannot be reversed, which is why it is very important to perform this form of data sampling with great care and detail, making sure that the data distribution of all the samples has not been greatly affected after data sampling.

Fog computing is an architecture that uses powerful fog devices to carry out a substantial amount of computation (*Nee & Nugroho, 2020*). This in turn eases computational load at centralized servers (*Afrin et al., 2019*; *Alam et al., 2019*; *D'Agostino et al., 2019*; *Martins et al., 2019*; *Xu et al., 2019*; *Zhao et al., 2019*). The fog computing layer lies typically between the centralized server layer and the smart devices layer as shown in Fig. 1. In the context of fog computing, discarding data before training begins is a much-needed solution because training neural networks on high dimensional data is a computational process that is only suitable for centralized servers as they are the ones that possess powerful hardware and storage capabilities. Therefore, fog devices must offload certain tasks to the server to meet latency and Quality-of-Service (QoS) requirements (*Baek & Kaddoum, 2021*). However, the transmission of high-dimensional data is an expensive process in terms of communication costs (*Eshratifar, Abrishami & Pedram , 2021*; *Eshratifar & Pedram, 2018*; *Liu et al., 2019*; *Song et al., 2018*). Therefore, the challenge in such scenarios is to discard a number of samples at the fog device which can reduce transmission cost from the fog device to the server along with accelerating the training time on the server while retaining the classification accuracies. As for the applications of fog computing for computer vision, two of the applications are in the domains of security surveillance and autonomous vehicles whereby in both cases, inferences on incoming images from cameras can be performed on the fog device such as object detection and classification and further analytics and/or training takes place on the server.

In the deep learning context, class incremental learning is about learning parameters for new incoming classes while retaining knowledge of the previously learned classes. The choice of training a deep learning model statically or incrementally really depends on the

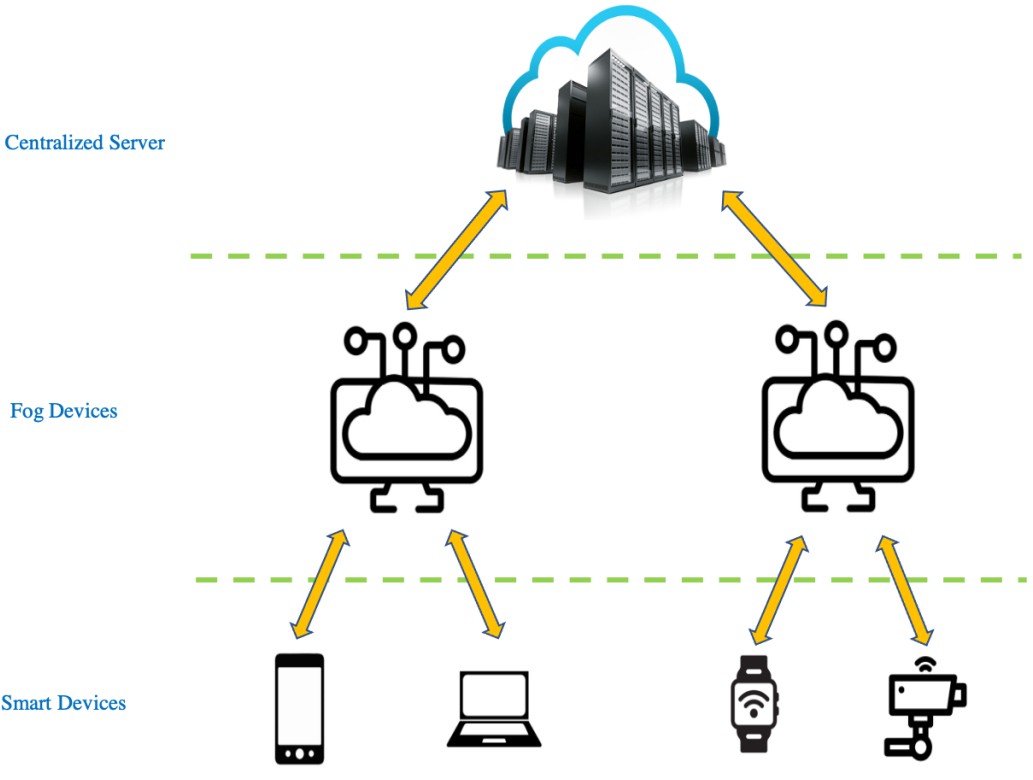

**Figure 1** Basic architecture of fog computing.

application being dealt with. This has led to state-of-the-art developments in incremental learning (*Choi, Lee & Choi, 2019*; *Gepperth & Karaoguz, 2016*; *Hayes et al., 2019*; *Kemker & Kanan, 2018*; *Lesort et al., 2020*; *Nallaperuma et al., 2019*; *Parisi et al., 2018*; *Rebuffi et al., 2017*; *Rusu et al., 2016*). All of these works focus on using techniques such as regularization, rehearsal, and dynamic network expansion to minimize catastrophic forgetting. The works that attempt to alleviate catastrophic forgetting via regularization do so by fixing the weights belonging to previous classes and/or preventing the weights belonging to old classes by changing much in an effort to retain old knowledge while learning new tasks. The works that attempt to alleviate catastrophic forgetting via rehearsal do so by using old samples to train the model along with the samples belonging to new tasks. The works that attempt to alleviate catastrophic forgetting via dynamic network expansion do so by increasing the network capacity (increasing the number of neurons and/or increasing the number of layers) in an effort to learn patterns of completely new incoming samples. Despite the fact that lots of data are needed for learning incrementally, the area of data sampling remains largely untapped in the field of incremental learning. Our data sampling algorithm is capable of sampling data from image classes before training even starts by automatically selecting a number of samples per class needed for learning. Our data sampling algorithm is able to perform data sampling for both static and incremental learning.

Data sampling becomes even more important when dealing with class incremental learning because, in class incremental learning, models need to be trained continuously at regular intervals as new data arrives and the computational and communication costs keep growing which is why data sampling must be performed on new incoming data to reduce the training and communication costs.

The contribution of this paper is a method for discarding a certain number of samples from the original training distribution of each class on the fog device (before training starts on the server) without affecting the model performance irrespective of the dataset, model, and learning settings. We apply this method in an image classification-based class incremental learning task to validate the findings.

The remaining of this article is presented in the following order: state-of-the-art research is discussed in 'Related Work'. Our system architecture and the algorithm behind our data sampling method are explained in 'Materials & Methods'. 'Experimental Settings' explains the software, hyperparameters, and learning settings used. We present and discuss our findings in 'Results' and 'Discussion', we then conclude our work in this article in the 'Conclusion' section.

## RELATED WORK

It is a known fact that not all the data out of a given training distribution is needed for training deep learning models. A lot of research has been done in this area that discards training data during training (*Alain et al., 2015*; *Gopal, 2016*; *Katharopoulos & Fleuret, 2018*; *Loshchilov & Hutter, 2015*; *Needell, Srebro & Ward, 2016*) whereby stochastic gradient descent is performed with importance sampling i.e., after forward propagation, only the useful samples are selected for backpropagation or by discarding samples whose loss values do not change much after a few epochs of training. However, all these approaches have one thing in common, these methods perform data sampling only after training has begun whereas our data sampling algorithm can discard a number of samples from a given training data distribution before the training even starts.

Recently, the work in *Birodkar, Mobahi & Bengio (2019)* reported that certain training samples in a training data distribution are not useful and can be discarded before training. The goal is achieved by discarding redundant samples from the training data distribution. This approach has two main problems. Firstly, the features of every image must be compared with every other feature in the dataset, which is highly computationally expensive especially for large datasets. Secondly, if in a training dataset, a particular class contains images of just faces of a single person then technically, most of the images might be deemed redundant by such algorithms and this would lead to a huge amount of data sampling causing a high amount of information loss.

There are several uncertainty sampling techniques (*Settles, 2009*) in active learning that are able to sample data based on query strategies. E.g., query strategies can select samples based on least confidence, highest loss, highest expected model change, etc. Though these query strategies have been very successful, these methods are not capable of counting how much data can be sampled from a training distribution i.e., sample size selection mechanism is not included. Therefore, an algorithm is needed that not only has a query

strategy but also has a mechanism to automatically discard a number of samples from a given training data distribution without affecting learning performance. This is exactly what we propose in this paper.

The approach in *Liu et al. (2019)* proposes a data sampling algorithm for an incremental learning image classification application. To do so, this work computes the confidence (accuracy) of the samples and if the confidence is smaller than a threshold, it indicates that such samples are hard to understand for the model and thus, such samples are selected for training the model further. However, the incremental learning model presented in *Liu et al. (2019)* can only learn new streaming examples of the same class. We go a step further and apply data sampling to an incremental learning model that learns completely new streaming classes. The challenge here is to perform data sampling on novel classes for which the model has no knowledge because the model has never seen such data before.

The works presented in *Liu et al. (2019)*, *Song et al. (2018)* carry out data sampling at fog devices before performing training on the server, however, none of these works carrying out pre-training data sampling on novel data classes. The works in (*López Chau, Li & Yu, 2013*; *Shen et al., 2016*; *Wang et al., 2013*) carry out pre-training data sampling and still manage to retain model performances, however, these works highlight pre-training data sampling for support vector machines and also do not consider data sampling for incremental learning. We aim to perform pre-training data sampling for artificial neural networks.

Instead of achieving state-of-the-art representation learning or improving classification accuracies, our work strictly focuses on proposing a data sampling algorithm with a dynamic sample size selection mechanism embedded in it which a lot of data sampling algorithms and even active learning strategies do not possess. To the best of our knowledge, what we propose is one of the first attempts of its kind since most of the notable work in class incremental learning focuses on achieving state-of-the-art classification accuracies (*Castro, Mar & Schmid, 2018*; *Kemker & Kanan, 2018*; *Rebuffi et al., 2017*; *Rusu et al., 2016*; *Wu et al., 2019*) but do not consider the possibility of carrying out incremental learning using a subsample of the overall training distribution which can reduce transmission costs to the server and the training time. However, along with that, our main priority is to maintain the classification accuracies during incremental learning.

## MATERIALS & METHODS

Figure 2 shows our system pipeline. We use a feature extractor of a Convolutional Neural Network (CNN) pre-trained on ImageNet (*Fei-Fei, Deng & Li, 2010*) for extracting features from the new incoming images. We employ transfer learning whereby we set the learning rate of CNN layers to be 0. This is because layers of CNNs pre-trained on ImageNet (*Fei-Fei, Deng & Li, 2010*) already produce features that are highly significant for classification. The way we implement transfer learning is by using pre-trained CNN feature extractors that are trained on ImageNet i.e., in incremental learning, new incoming images are passed through this pre-trained CNN feature extraction after which these 'transferred features' are sent to the server where the classifier (Fully connected layers) is trained from scratch. The reason

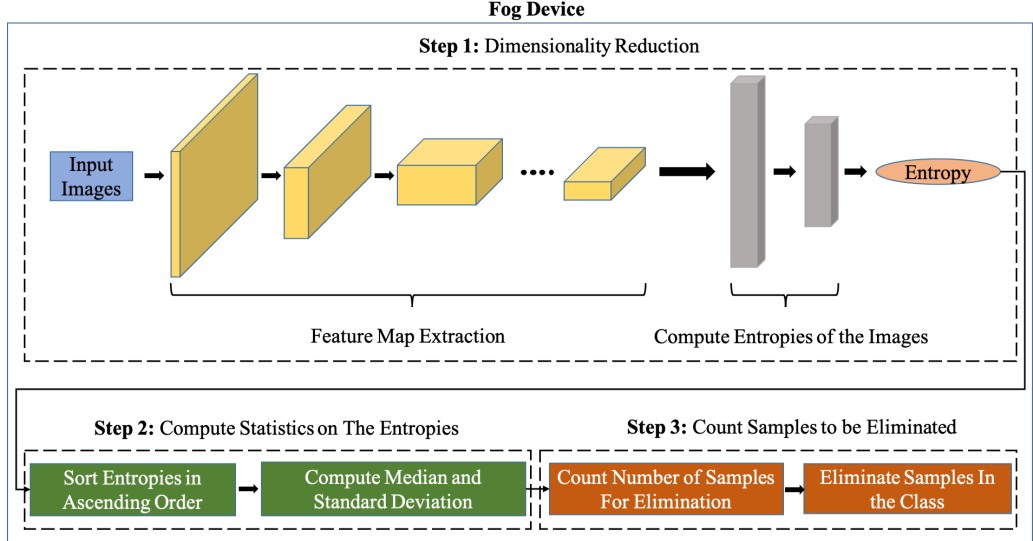

**Figure 2** **Overview of the Sample Elimination Algorithm on a fog device.**

why we freeze the convolutional layers of the pre-trained CNNs is to avoid performing backpropagation on the fog device for every mini-batch in the training dataset, this way, the training can take place only on the server without transmitting the backpropagated gradients of the classifier on the server back to the fog device for every mini-batch.

It can be seen from Fig. 3 that the feature extractor (CNN layers of the pre-trained CNN) is run on the fog device and an Artificial Neural Network (ANN) based classifier is run on the server. The pre-trained feature extractors are used to extract meaningful features from input images and the classifier on the server is a neural network that is trained on incoming data samples. We freeze the feature extractor which means that the CNN layers will not be trained because, once backpropagation has taken place on the server, the gradients of the classifier must then be sent back to the fog device after which backpropagation of the CNN layers of the feature extractor must be carried out on the fog device if the CNN layers are not frozen. This leads to a huge communication cost between the fog device and the server along with a very expensive computation cost associated with performing backpropagation on the feature extractor at the fog device.

We perform data sampling for each class. When samples from novel classes start arriving, they are forward propagated through the convolutional layers of a CNN pre-trained on ImageNet (*Fei-Fei, Deng & Li, 2010*). Our proposed Sample Elimination Algorithm (SEA) takes place after this whereby the features of the images are forward propagated through an ANN which is a single layer artificial neural network having $\beta$ output neurons where $\beta$ is the total classes per incremental training round that will be learned, we let this ANN be denoted as $ANN_{fog}$. The weights of $ANN_{fog}$ are always randomly initialized for the new $\beta$ classes to be learned. By using $ANN_{fog}$, we obtain the entropies of images. We then use the entropy distribution of each class and compute the sample size to be discarded from each class. $ANN_{fog}$ is never trained and is only used for computing entropies of images for

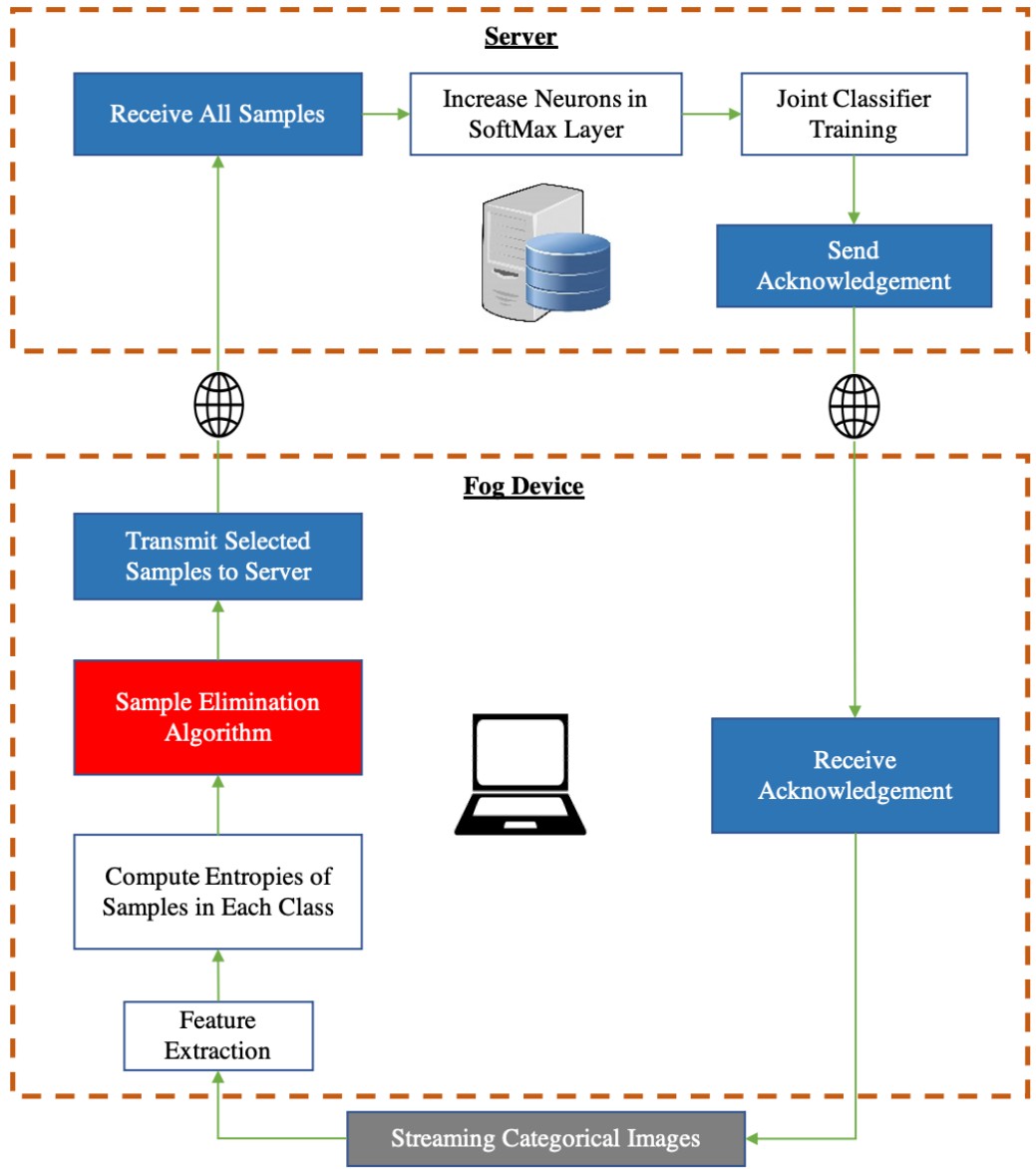

**Figure 3 Incremental learning system split between a fog device and a centralized server.** Red block: our contribution. Gray block: Input data. Blue block: In-built software communication modules.

data sampling. The reason why the weights of $ANN_{fog}$ are always randomly initialized is because of new $\beta$ neurons being added to the classifier on the server after every incremental training round for learning new tasks hence, the weights of $ANN_{fog}$ must also be initialized in a randomized manner.

Once data sampling has taken place for each class, the features of the selected samples are transmitted to the server. In class incremental learning, it is possible to learn a few classes at a time which is why a particular incremental training round can have a few classes being trained together. E.g., if we wish to train our neural network after every 10 classes

have arrived, then each incremental training round will contain 10 classes. Therefore, once all samples from all classes in an incremental training round have been forward propagated through the CNN feature extractor and $ANN_{fog}$, we finally start the main incremental learning process on the server.

Certain datasets are imbalanced in nature and after SEA is applied to every class, the number of samples in each class may vary as well which is why we balance the number of samples in each class by applying the Synthetic Minority Over-sampling Technique (SMOTE) (*Chawla et al., 2002*). As we have already mentioned that we are not contributing any novel core incremental learning algorithm, we simply carry out incremental learning by performing joint training which is one of the most common incremental learning techniques that combines both the new samples with the old samples and trains the neural network as the incremental learning progresses.

Once $ANN_{server}$ has finished learning from an incremental training round, the SoftMax classification layer of $ANN_{server}$ is modified i.e., $\beta$ new neurons are added to this layer. Each neuron is responsible for classifying one of the total classes that have been learned so far and since we are incrementally learning $\beta$ classes at a time, we add $\beta$ neurons to the SoftMax classification layer, and only the weights associated with these neurons are initialized in a random manner for learning from new classes.

We now describe the details of SEA. SEA attempts to discard a certain number of samples per class. Let $c$ be the index number of a class in a dataset and $n_c$ be the total samples in class $c$. Let $E^c$ be a set that holds all the entropies/losses for all samples in class $c$ such that $E^c \in e_0^c, e_1^c, e_2^c, \ldots, e_{n_c}^c$ where $E^c$ is sorted in ascending order. Entropies in $E^c$ are obtained via the cross-entropy cost objective as shown in Eq. (1).

$$e_j^c = \left| \overrightarrow{\left(l_j^c\right)^T} \cdot \overrightarrow{log_e \varphi\left(\left(W_{fog}' \cdot x_j^c\right) + b_{fog}'\right)} \right| \tag{1}$$

In Eq. (1), the formula for cross-entropy loss is shown. The notation $(\cdot)$ represents the dot product, $x_j^c$ is the output of the convolutional layers of a pre-trained CNN for sample $j$ from class $c$, $\overrightarrow{\left(l_j^c\right)}^T$ is the label (one-hot label vector) corresponding to $x_j^c$. $W_{fog}'$ and $b_{fog}'$ are the weights and biases of the new $\beta$ neurons of $ANN_{fog}$. Since there is no precise mechanism to predict how the images from novel classes will affect the training of $ANN_{server}$ or any other neural network for that matter, therefore $W_{fog}'$ and $b_{fog}'$ are randomly initialized. $\varphi(\cdot)$ is the SoftMax function. $e_j^c$ represents the entropy of sample $j$ of set $E^c$.

$$\sigma_E^c = \left(\frac{\sum_{j=1}^{n_c}(e_j^c - \mu_E^c)^2}{n_c}\right)^{\frac{1}{2}} \tag{2}$$

In Eq. (2), the standard deviation of $E^c$ is computed where $\sigma_E^c$ is the standard deviation of the set $E^c$ and $\mu_E^c$ is the median of the set $E^c$. The reason why we choose the median instead of the mean is to avoid the problem of entropy skewness in $E^c$, e.g., if the mean is used and if one of the samples in $E^c$ has a very high entropy value, the mean of the entire list can increase greatly and can cause unstable data sampling which can greatly affect the

incremental learning process. However, choosing the median of $E^c$ will not be affected by such skewed entropies.

$$\omega^c = \arg\max_{\forall e_j^c \in E^c}(e_j^c \cdot \mathbb{I}\left(\mu_E^c - e_j^c < \sigma_E^c\right)) \qquad (3)$$

In Eq. (3), the notation $(\cdot)$ represents the scalar product. $\mathbb{I}(\cdot)$ is the indicator function whose output is one if the condition $\left(e_0^c \leq e_j^c < \omega^c\right)$ is satisfied, if this condition is not satisfied then the output of $\mathbb{I}(\cdot)$ is zero. The term $\omega^c$ represents the cut-off loss and its role is to count all samples with entropies smaller than $\omega^c$.

$$\rho^c = \sum_{j=1}^{n_c} \mathbb{I}\left(e_0^c \leq e_j^c < \omega^c\right) \qquad (4)$$

In Eq. (4), $\rho^c$ denotes the total number of samples that can be discarded from class $c$ before the remaining $n_c - \rho^c$ samples are transmitted to the server. In Eq. (4), we keep counting the number of samples as long as the condition inside the indicator function is being satisfied.

To summarize our proposed SEA, the training samples in each class are sorted with respect to their loss values in ascending order. This loss distribution of each class is used to perform data sampling. We then compute the standard deviation of this loss distribution as we want to know by how much loss values differ from the median of this distribution. We compute the difference between loss values of $E^c$ and $\mu_E^c$ and compare the difference with $\sigma_E^c$, we start computing the difference starting from the lowest loss value in $E^c$. We then take the maximum loss value in $E^c$ that satisfies (3). We call this loss value the cut-off loss that is denoted by $\omega^c$. We then arrive at the final stage of our proposed SEA i.e., to determine the sample size to be discarded per class. We count all samples that have entropy values that lie between $e_0^c$ (minimum loss value) and $\omega^c$ (cut-off loss value) in $E^c$.

The intuition behind SEA is to select samples with entropies that vary highly from the median of the loss distribution of the samples per class. This is because samples with high entropies are the ones that a deep learning model does not understand thus, these samples must be trained further since such samples are considered hard samples. This is why we compute the standard deviation of the loss distribution, and since the standard deviation of a list represents by how much all the samples in the list deviate from the median, we compute the difference of low loss samples from the median and count how many such samples have a difference which is even smaller than the standard deviation, this indicates that such samples have a very low deviation from the median of the entropy distribution of the class and will not contribute greatly to the ANN parameter learning at the server. Hence, these are the total number of samples that are not needed for training and are discarded at the fog device. If the loss value of the very first sample in a batch of $n_c$ images does not satisfy (3), this implies that $\sigma_E^c$ has a very low value meaning that there is not much deviation in the data distribution of $n_c$ images and hence $\omega^c$ is set to the minimum loss value ($e_0^c$) which means that no data sampling takes place for class $c$. However, if $\rho^c$ is greater than zero, then the total samples to be eliminated ($\rho^c$) from class $c$ are then discarded randomly. This is done because of the random weight initialization of newly

**Table 1  Learning settings and datasets used for all experiments.**

| | Datasets | | | |
|---|---|---|---|---|
| Learning settings | CUB-200 (*Welinder et al., 2010*) | Stanford dogs (*Khosla et al., 2011*) | CIFAR-100 (*Krizhevsky, 2009*) | ImageNet (*Fei-Fei, Deng & Li, 2010*) |
| Training epochs | 25 | 30 | 15 | 3 |
| Number of fully connected layers | 1 | 1 | 1 | 2 |
| Dropout probability | 0.25 | 0.30 | 0.20 | 0.30,0.20 |
| Learning rate | 0.0015 | 0.0002 | 0.0001 | 0.0035 |
| Batch size | 16 | 32 | 64 | 128 |
| Number of classes learnt at a time | 50 | 40 | 20 | 2 |
| Image size | $90 \times 90 \times 3$ | $85 \times 85 \times 3$ | $32 \times 32 \times 3$ | $100 \times 100 \times 3$ |
| CNN model | SqueezeNet (*Iandola et al., 2016*) | MobileNetV2 (*Sandler et al., 2018*) | MnasNet (*Tan et al., 2019*) | ShuffleNet V2 (*Ma et al., 2018*) |
| Train/test split | 5994/5794 | 12000/8580 | 50000/10000 | 6937/1728 |
| Train/Test split (%) | 50.84/49.16 | 58.30/41.70 | 83.30/16.60 | 80.00/20.00 |

assigned $\beta$ neurons of $ANN_{server}$ and because of the stochasticity of neural networks, it is possible for any sample to have a final loss value after training that is greater than its own initial loss value before the start of the training. In order to reduce the frequency of such samples that exhibit such behavior, we perform data sampling in a randomized manner. This is discussed in much detail in Results and Discussion.

## Experimental settings

We now describe the details of our experiment implementations to evaluate the performance of our proposed data sampling algorithm (SEA). Table 1 shows the datasets, models, and hyperparameters used for our experiments. Apart from CIFAR-100 (*Krizhevsky, 2009*), all the datasets are imbalanced in nature. We only choose RGB images present in each of these datasets and all the images in these datasets are resized before training begins as shown in Table 1. All the hyperparameters have been randomly chosen to show that SEA can perform under any learning setting. Achieving the state-of-the-art accuracy for incremental learning is not our goal thus the hyperparameters shown in Table 1 have not been fine-tuned. We would like to point out that when working with ImageNet (*Fei-Fei, Deng & Li, 2010*), we use two fully connected layers in $ANN_{server}$ therefore, for the first layer which has 1000 neurons, we use a dropout probability of 0.30 and for the second and the final layer (SoftMax classification layer) of $ANN_{server}$, we use a dropout probability of 0.20. Since we are dealing in a fog computing scenario, the CNN models must be lightweights in nature hence, we choose four different lightweights CNN models which are as follows: SqueezeNet (*Iandola et al., 2016*), MobileNetV2 (*Sandler et al., 2018*), MnasNet (*Tan et al., 2019*), and ShuffleNetV2 (*Ma et al., 2018*). Another reason for choosing four different CNNs is to ensure the robustness of SEA under various CNN architectures.

As mentioned, SEA counts the number of samples to be discarded per class and discards these samples in a randomized manner. We compare SEA with four baseline methods: No

Sampling (NS) and three non-parametric statistical methods: Wilcoxon Rank-Sum Test (WRST) (*Ozcan & Basturk, 2020*), Median Test (MT) (*Alippi, Boracchi & Roveri, 2017*), and Kruskal–Wallis Test (KWT) (*Höpken et al., 2020*). Firstly, WRST (*Ozcan & Basturk, 2020*), MT (*Alippi, Boracchi & Roveri, 2017*), and KWT (*Höpken et al., 2020*) are not built on top of SEA and secondly, these methods are non-parametric in nature that do not assume any data distribution in advance. Furthermore, these methods are very strong in comparing statistical differences between two groups whereas KWT (*Höpken et al., 2020*) can compare statistical differences across multiple groups. Data sampling leads to the formation of two groups i.e., one group containing all samples in a class ($n_c$) and the second group containing only the selected samples per class for training ($n_c - \rho^c$) which is why these non-parametric methods are ideal for comparison with SEA. We apply WRST (*Ozcan & Basturk, 2020*), MT (*Alippi, Boracchi & Roveri, 2017*), and KWT (*Höpken et al., 2020*) to $E^c$ for performing data sampling. The formulas to compute WRST (*Ozcan & Basturk, 2020*), KWT (*Höpken et al., 2020*), and MT (*Alippi, Boracchi & Roveri, 2017*) are shown in Eqs. (5)–(9), (10)–(12), and (13)–(19) respectively.

$$\mu_w = \frac{n_A \cdot (n_A + n_B + 1)}{2} \tag{5}$$

$$\sigma_w = \sqrt{\frac{n_A \cdot n_B (n_A + n_B + 1)}{12}} \tag{6}$$

$$z = \frac{R - \mu_w}{\sigma_w} \tag{7}$$

$$p = \frac{1}{2} \cdot \left[1 + \mathrm{erf}\left(\frac{z}{\sqrt{2}}\right)\right] \tag{8}$$

$$\vartheta = \begin{cases} 1, p < \alpha \\ 0, otherwise \end{cases} \tag{9}$$

In Eqs. (5)– (9), $n_A$ and $n_B$ represent the number of elements in data distributions $A$ and $B$, respectively. $\mu_w$ and $\sigma_w$ denote the mean and the standard deviation, respectively. $R$ is the rank sum of the elements in data distribution $A.z$ represents the $z$ score and $p$ represents the $p$-value. $\alpha$ is the statistical significance value which is set to 0.05 in our experiments.

$$n_T = \sum_{i=1}^{k} n_i \tag{10}$$

$$H = \left[\frac{12}{n_T \cdot (n_T + 1)} \cdot \sum_{i=1}^{k} \frac{R_i^2}{n_i}\right] - 3 \cdot (n_T + 1) \tag{11}$$

$$\vartheta = \begin{cases} 1, H > X^2 \\ 0, otherwise \end{cases} \tag{12}$$

In Eqs. (10)–(12), $k$ represents the total data distributions that are being compared, $k$ is set to two in our case since we are comparing two data distributions i.e., the distribution with all entropy values of the class being processed and the distribution with the selected

entropy values of the class being processed. $n_i$ is the total samples in a data distribution $k$ and $n_T$ is the total samples across all the data distributions. $R_i$ is the rank sum of all elements in data distribution $k$. $X^2$ is the inverse of the $p$-value for a chi-square distribution where $p$ is set to 0.05 in our experiments.

$$W = \sum_{i=1}^{n_A} \mathbb{I}(A_i > \widetilde{\mu_{AB}}) \qquad (13)$$

$$X = \sum_{i=1}^{n_B} \mathbb{I}(B_i > \widetilde{\mu_{AB}}) \qquad (14)$$

$$Y = \sum_{i=1}^{n_A} \mathbb{I}(A_i < \widetilde{\mu_{AB}}) \qquad (15)$$

$$Z = \sum_{i=1}^{n_B} \mathbb{I}(B_i < \widetilde{\mu_{AB}}) \qquad (16)$$

$$N_{AB} = W + X + Y + Z \qquad (17)$$

$$X^2 = \frac{N_{AB} \cdot \left(W \cdot Z - X \cdot Y - \frac{N_{AB}}{2}\right)^2}{(W+X) \cdot (Y+Z) \cdot (W+Y) \cdot (X+Z)} \qquad (18)$$

$$\vartheta = \begin{cases} 1, X^2 > X_{\alpha, k-1}^2 \\ 0, otherwise \end{cases} \qquad (19)$$

In Eqs. (13)– (19), $\widetilde{\mu_{AB}}$ is the median of all the elements in the distributions $A$ and $B$ combined. $W$ and $X$ are the number of elements in the distributions $A$ and $B$ respectively that are greater than $\widetilde{\mu_{AB}}$. $Y$ and $Z$ are the number of elements in the distributions $A$ and $B$ respectively that are less than $\widetilde{\mu_{AB}}$. $N_{AB}$ is the total number of elements in distributions $A$ and $B$ combined. $X^2$ is the chi-square value. $X_{\alpha, k-1}^2$ is the chi-square value at significance level $\alpha$, $k-1$ is the degree of freedom where $k$ is the number of data distributions. $\alpha$ is set to 0.005.

In all non-parametric tests described above, if $\vartheta$ is 1, it means there is a statistical difference that is detected between distributions $A$ and $B$ thus all the samples corresponding to the entropies in distribution A are transmitted to the server. If $\vartheta$ is 0, it means there is not a statistical difference between distributions $A$ and $B$.

SMOTE (*Chawla et al., 2002*) is used to create new samples per class to tackle the problem of class imbalance. In Eq. (20), the formula to generate a new sample (*Chawla et al., 2002*) is shown.

$$p' = p_1 + \left[\lambda \cdot (p_2 - p_1)\right] \qquad (20)$$

In Eq. (20), $p'$ is the newly created point (sample). $p_1$ and $p_2$ are any two random points in the dataset. $\lambda$ is a uniformly random generated number between zero and one i.e., $\lambda \sim U([0,1])$. In our experiments, we randomly choose $p_1$ and $p_2$ per class to balance the number of samples across all classes.

All classification accuracies reported in our experiments are the top-5 accuracies (rounded to two decimal places). Each experiment is run three times and all metrics

**Table 2** Performance of the incremental learning system with respect to various data sampling algorithms when evaluating CIFAR-100 (*Krizhevsky, 2009*) using MnasNet (*Tan et al., 2019*).

| Data Sampling Algorithm | Number of Classes Trained | | | | |
|---|---|---|---|---|---|
| | 20 | 40 | 60 | 80 | 100 |
| (i) **Samples Needed for Training** | | | | | |
| NS | 10000 | 10000 | 10000 | 10000 | 10000 |
| **SEA** | **8578** | **8461** | **8406** | **8453** | **8485** |
| WRST (*Ozcan & Basturk, 2020*) | 9379 | 9393 | 9394 | 9393 | 9395 |
| MT (*Alippi, Boracchi & Roveri, 2017*) | 8742 | 8751 | 8759 | 8767 | 8757 |
| KWT (*Höpken et al., 2020*) | 9260 | 9267 | 9279 | 9272 | 9271 |
| (ii) **Training time (s)** | | | | | |
| NS | 4.36 | 8.38 | 12.32 | 16.48 | 20.50 |
| SEA | **3.49** | 7.22 | **10.84** | **13.91** | **17.80** |
| WRST (*Ozcan & Basturk, 2020*) | 4.15 | 8.02 | 11.81 | 15.29 | 18.90 |
| MT (*Alippi, Boracchi & Roveri, 2017*) | 3.87 | **7.18** | 10.95 | 14.22 | 18.30 |
| KWT (*Höpken et al., 2020*) | 3.94 | 7.76 | 11.99 | 16.21 | 19.51 |
| iii) **Classification Accuracy (%)** | | | | | |
| NS | 85.35 | 75.25 | 69.73 | 66.66 | 64.08 |
| SEA | **84.75** | **74.25** | 69.01 | 65.93 | **63.59** |
| WRST (*Ozcan & Basturk, 2020*) | 84.74 | 74.23 | **69.15** | **66.36** | 63.58 |
| MT (*Alippi, Boracchi & Roveri, 2017*) | 84.50 | 74.13 | 68.55 | 65.26 | 62.67 |
| KWT (*Höpken et al., 2020*) | 84.55 | 74.18 | 69.08 | 66.16 | 63.39 |

reported in our experiments are then averaged over these three runs. However, we report top-1 accuracy when working with the ImageNet (*Fei-Fei, Deng & Li, 2010*) dataset. We calculate the classification accuracies using the test set of each dataset. The PyTorch (*Paszke et al., 2019*) library is used to program the experiments. All of the input images are normalized by converting from a range of 0 - 255 to 0 - 1. The Nvidia Tesla K80 GPU is used as the server and a desktop computer with an i7 processor is used as a fog device to carry out all the experiments.

## RESULTS

In this section, we present and discuss our results. Tables 2–5 show the number of samples selected before incremental training starts, the time taken to train $ANN_{server}$, and the classification accuracies at every incremental training round. This is with respect to the learning settings shown in Table 1 using all the data sampling algorithms along with no data sampling. The numerical values in bold are the ones that corresponding to the best data sampling algorithm out of SEA, WRST (*Ozcan & Basturk, 2020*), MT (*Alippi, Boracchi & Roveri, 2017*), and KWT (*Höpken et al., 2020*) in terms of a given metric.

For CIFAR-100 (*Krizhevsky, 2009*), regardless of the data sampling algorithm, the classification accuracies obtained at every incremental training round are very similar to the ones obtained without any data sampling. However, the differences are observed in the number of samples needed for training and the training time at every incremental training round. SEA requires the least number of training samples and the least amount of training

**Table 3** Performance of the incremental learning system with respect to various data sampling algorithms when evaluating CUB-200 (*Welinder et al., 2010*) using SqueezeNet (*Iandola et al., 2016*).

| Data Sampling Algorithm | Number of Classes Trained | | | |
|---|---|---|---|---|
| | 50 | 100 | 150 | 200 |
| (i) **Samples Needed for Training** | | | | |
| NS | 1498 | 1495 | 1500 | 1498 |
| SEA | 1381 | 1345 | 1347 | 1346 |
| WRST (*Ozcan & Basturk, 2020*) | 1068 | 1066 | 1070 | 1069 |
| MT (*Alippi, Boracchi & Roveri, 2017*) | **723** | **720** | **725** | **717** |
| KWT (*Höpken et al., 2020*) | 986 | 974 | 979 | 979 |
| (ii) **Training Time (s)** | | | | |
| NS | 4.16 | 8.33 | 12.49 | 16.69 |
| SEA | 3.74 | 7.45 | 11.05 | 14.85 |
| WRST (*Ozcan & Basturk, 2020*) | 2.92 | 5.75 | 8.61 | 11.70 |
| MT (*Alippi, Boracchi & Roveri, 2017*) | **1.99** | **3.88** | **5.82** | **7.92** |
| KWT (*Höpken et al., 2020*) | 2.70 | 5.34 | 7.95 | 10.74 |
| (iii) **Classification Accuracy (%)** | | | | |
| NS | 51.48 | 43.36 | 33.09 | 29.67 |
| SEA | **49.67** | **40.88** | **32.05** | **27.77** |
| WRST (*Ozcan & Basturk, 2020*) | 47.95 | 37.37 | 30.82 | 25.51 |
| MT (*Alippi, Boracchi & Roveri, 2017*) | 44.56 | 32.54 | 25.80 | 20.85 |
| KWT (*Höpken et al., 2020*) | 47.59 | 35.06 | 28.83 | 24.59 |

time with the exception of the training time taken when learning 40 classes using MT (*Alippi, Boracchi & Roveri, 2017*). Overall, SEA performs better than other data sampling algorithms when evaluated on this dataset.

The CUB-200 (*Welinder et al., 2010*) dataset has a total of 200 classes but does not have a fixed number of images for every class. However, the number of images per class is very small which is the main reason why we choose this dataset to test SEA. Discarding information from a small data pool can lead to a big information change in the overall data pool which is why we must test SEA to really observe its effectiveness.

For CUB-200 (*Welinder et al., 2010*), SEA is able to reject the least samples as compared to other data sampling algorithms whereby the least samples needed for training arise when using MT (*Alippi, Boracchi & Roveri, 2017*). However, the classification accuracies obtained using SEA are the best out of all the data sampling methods. Furthermore, the classification accuracies obtained at every incremental training round are within 3% of the baseline (NS). The fact that WRST (*Ozcan & Basturk, 2020*), MT (*Alippi, Boracchi & Roveri, 2017*), and KWT (*Höpken et al., 2020*) discard so many samples indicates that these statistical methods are only able to conclude a significant difference between the overall metric distribution (all samples per class) and the selected distribution (samples to be transmitted to the server) once a high number of samples are eliminated. Clearly, for datasets with a small number of samples per class such as CUB-200 (*Welinder et al., 2010*), SEA is much better suited than other data sampling methods.

**Table 4  Performance of the incremental learning system with respect to various data sampling algorithms when evaluating Stanford Dogs (*Khosla et al., 2011*) using MobileNetV2 (*Sandler et al., 2018*).**

| Data Sampling Algorithm | Number of Classes Trained | | |
|---|---|---|---|
| | 40 | 80 | 120 |
| **(i) Samples Needed for Training** | | | |
| NS | 4000 | 3999 | 4000 |
| SEA | 3505 | 3440 | 3477 |
| WRST (*Ozcan & Basturk, 2020*) | 3403 | 3410 | 3411 |
| MT (*Alippi, Boracchi & Roveri, 2017*) | **2879** | **2877** | **2871** |
| KWT (*Höpken et al., 2020*) | 3318 | 3319 | 3316 |
| **(ii) Training Time (s)** | | | |
| NS | 8.93 | 17.13 | 25.09 |
| SEA | 7.77 | 15.43 | 22.15 |
| WRST (*Ozcan & Basturk, 2020*) | 7.68 | 15.29 | 22.14 |
| MT (*Alippi, Boracchi & Roveri, 2017*) | **6.45** | **12.78** | **18.96** |
| KWT (*Höpken et al., 2020*) | 7.39 | 14.61 | 21.32 |
| **(iii) Classification Accuracy (%)** | | | |
| NS | 62.89 | 52.13 | 44.78 |
| SEA | 61.78 | **51.66** | **43.17** |
| WRST (*Ozcan & Basturk, 2020*) | 62.11 | 49.99 | 43.14 |
| MT (*Alippi, Boracchi & Roveri, 2017*) | 60.88 | 48.96 | 41.27 |
| KWT (*Höpken et al., 2020*) | **62.39** | 51.27 | 43.16 |

Stanford Dogs (*Khosla et al., 2011*) dataset has a total of 120 classes where each class has a different number of samples. However, the number of samples in almost every class exceeds 100 images. For Stanford Dogs (*Khosla et al., 2011*), the number of samples needed for training and the training time is the least when using MT (*Alippi, Boracchi & Roveri, 2017*). This again indicates that MT (*Alippi, Boracchi & Roveri, 2017*) detects a significant difference between the overall metric distribution (all samples per class) and the selected distribution (samples to be transmitted to the server) once a high number of samples are eliminated. However, there is not a huge degradation in the classification accuracies as compared to what was observed when evaluating CUB-200 (*Welinder et al., 2010*), this is because the number of samples per class in Stanford Dogs (*Khosla et al., 2011*) is a lot more as compared to CUB-200 (*Welinder et al., 2010*). However, MT (*Alippi, Boracchi & Roveri, 2017*) does not achieve the highest classification accuracy at even a single incremental training round. SEA leads to the highest classification accuracies obtained at two out of three incremental training rounds.

ImageNet (*Fei-Fei, Deng & Li, 2010*) is a well know large-scale dataset which is why it is very important to apply and test data sampling algorithms on a dataset that has a large number of samples per class. ImageNet (*Fei-Fei, Deng & Li, 2010*) has a very large number of classes, however, since we are more interested to see how data sampling algorithms perform on classes with a large number of samples, we only use a total of six classes from ImageNet (*Fei-Fei, Deng & Li, 2010*). However, these six classes are among the top 10 classes with the most number of samples in a class in the ImageNet (*Fei-Fei, Deng & Li,*

**Table 5** Performance of the incremental learning system with respect to various data sampling algorithms when evaluating ImageNet (*Fei-Fei, Deng & Li, 2010*) using ShuffleNetV2 (*Ma et al., 2018*).

| Data Sampling Algorithm | Number of Classes Trained | | |
|---|---|---|---|
| | **2** | **4** | **6** |
| (i) **Samples Needed for Training** | | | |
| NS | 1868 | 2435 | 2631 |
| SEA | **1540** | **2066** | **2223** |
| WRST (*Ozcan & Basturk, 2020*) | 1785 | 2340 | 2533 |
| MT (*Alippi, Boracchi & Roveri, 2017*) | 1698 | 2246 | 2436 |
| KWT (*Höpken et al., 2020*) | 1767 | 2322 | 2513 |
| ((ii) **Training Time (s)** | | | |
| NS | 1.00 | 2.04 | 3.01 |
| SEA | **0.64** | **1.75** | **2.51** |
| WRST (*Ozcan & Basturk, 2020*) | 0.70 | 1.97 | 2.91 |
| MT (*Alippi, Boracchi & Roveri, 2017*) | 0.69 | 1.91 | 2.75 |
| KWT (*Höpken et al., 2020*) | 0.70 | 1.97 | 2.86 |
| (iii) **Classification Accuracy (%)** | | | |
| **NS** | 97.01 | 95.07 | 86.37 |
| SEA | 97.22 | **95.45** | 85.27 |
| WRST (*Ozcan & Basturk, 2020*) | 97.01 | 94.80 | 83.48 |
| MT (*Alippi, Boracchi & Roveri, 2017*) | 97.22 | 94.42 | **86.83** |
| KWT (*Höpken et al., 2020*) | **97.44** | 93.31 | 83.71 |

*2010*) dataset. The classes we use are as follows: n01882714 (koala bear, kangaroo bear, native bear), n02086240 (Shih-Tzu), n02094433 (Yorkshire terrier), n02138441 (meerkat), n02279972 (monarch butterfly, Danaus plexippus), and n09428293 (seashore).

For ImageNet (*Fei-Fei, Deng & Li, 2010*), SEA manages to utilize the least number of training samples and the least time for training as compared to all the other data sampling methods. Furthermore, all classification accuracies obtained irrespective of the data sampling algorithm are within 3% of the baseline accuracies therefore, SEA is the better performing algorithm in this experiment. The reason why the training time is so small is that we are only experimenting with six classes of ImageNet (*Fei-Fei, Deng & Li, 2010*) and the training epochs are very small thus there is not a very high training time. Despite reporting top-1 accuracies when working on ImageNet (*Fei-Fei, Deng & Li, 2010*) dataset, the classification accuracies that we obtain at every incremental training round are very high and this is because all the models mentioned in Table 1 are already pre-trained on this dataset.

From Tables 2–5 we showed that SEA is the best performing data sampling algorithm, however, we must further examine the performance and effectiveness of SEA. Table 6 shows the standard deviation in the number of samples transmitted to the server with respect to NS, time taken for training on the server, and the classification accuracies all of

**Table 6** Standard deviation of the reduction in the number of training samples, reduction in the training time on the server and the difference in classification accuracies after applying SEA.

| Datasets | Standard deviations between NS and SEA | | |
|---|---|---|---|
| | Samples transmitted | Training time (s) | Accuracies (%) |
| CIFAR-100 (*Krizhevsky, 2009*) | 1077 | 15.87 | 0.50 |
| CUB-200 (*Welinder et al., 2010*) | 101 | 5.32 | 1.27 |
| Stanford Dogs (*Khosla et al., 2011*) | 1115 | 4.10 | 0.75 |
| ImageNet (*Fei-Fei, Deng & Li, 2010*) | 781 | 0.82 | 0.40 |

which are calculated using formulas shown in Eqs. (21)–(23) respectively.

$$\text{Samples}_{\text{dev}} = \left\lfloor \frac{\left| \sum_{t=1}^{T} \text{Samples}_{\text{SEA}}^{t} - \sum_{t=1}^{T} \text{Samples}_{\text{NS}}^{t} \right|}{\sqrt{2}} \right\rfloor \tag{21}$$

$$\text{Train}_{\text{dev}} = \frac{\left| \sum_{t=1}^{T} \text{Train}_{\text{SEA}}^{t} - \sum_{t=1}^{T} \text{Train}_{\text{NS}}^{t} \right|}{\sqrt{2}} \tag{22}$$

$$\text{Acc}_{\text{dev}} = \frac{1}{T} \cdot \sum_{t=1}^{T} \frac{\left| \text{Acc}_{\text{SEA}}^{t} - \text{Acc}_{\text{NS}}^{t} \right|}{\sqrt{2}} \tag{23}$$

In Eqs. (21)–(23), $T$ represents the total number of incremental training rounds. At incremental training round $t$, $Samples_{SEA}^{t}$ denotes the number of samples transmitted from the fog device to the server, $Train_{SEA}^{t}$ represents the training time of $ANN_{server}$, $Acc_{SEA}^{t}$ represents the classification accuracy. In Eqs. (21) and (22), using SEA and NS, we compute the standard deviation in the total samples sent to the server and the total training time of $ANN_{server}$, respectively. We then compute the standard deviation between SEA and NS for the metrics: total samples transmitted to the server and the total training time on the server. At each incremental training round, we obtain a classification accuracy, therefore, in Eq. (23), the standard deviation of the classification accuracy between SEA and NS is computed at each incremental training round, and then the average of all standard deviations is computed.

It is shown in Table 6 that the standard deviation of the classification accuracies is less than one for all datasets except CUB-200 (*Welinder et al., 2010*) which also has a very small standard deviation of 1.27 even though the total samples per class in this dataset is very low. For such small standard deviations in the classification accuracies, we achieve a very large standard deviation in the reduction of the number of training samples and also a decent standard deviation in the reduction of the training time after applying SEA.

The learning settings for evaluating each dataset are different which is why the standard deviations in the training time reduction are different for each dataset i.e., not directly proportional to the reduction in the number of training samples. However, the main point here is that the results in Table 6 support our hypothesis that after using SEA, the classification accuracies are very similar to the ones obtained without data sampling while reducing the training time and the number of samples transmitted to the server.

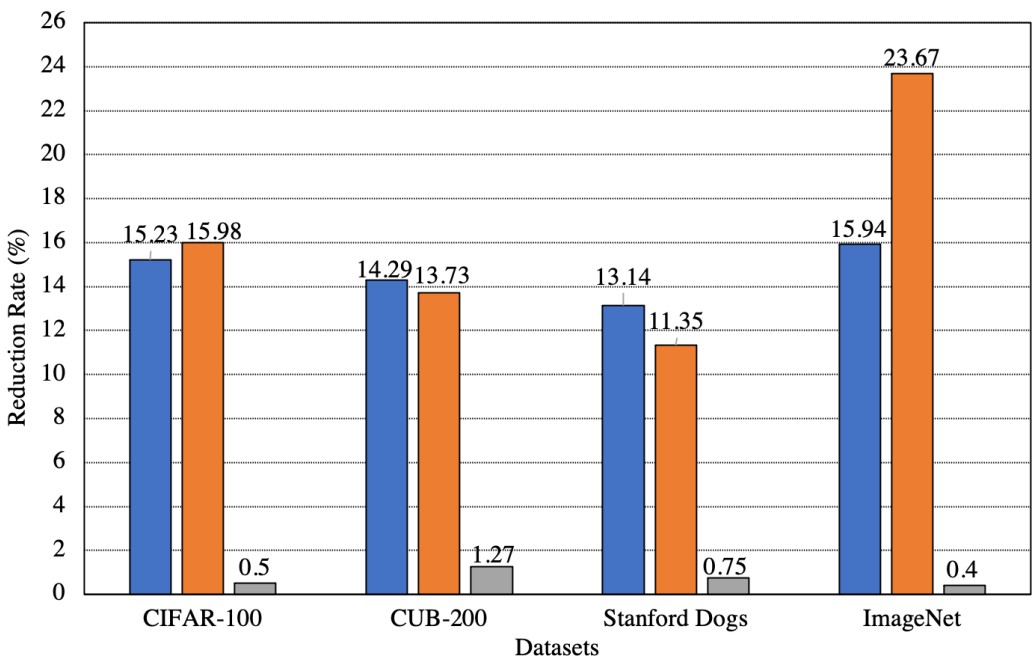

**Figure 4** **Overall change in the number of samples transmitted to the server, the training time on the server, and the classification accuracies after applying SEA.** Blue: Samples transmitted. Orange: Training time on server. Gray: Accuracy.

We train our model incrementally by carrying out joint training and by doing so, we also show that SEA can also work for static training. In joint training, all the samples of the new classes are concatenated together with all samples of the old classes, and the model is then trained. However, if we observe closely, joint training is simply a series of static training sessions. E.g., if we look at the first row of Tables 2–5, at this stage, no training has taken place but when the training begins, the model is trained on all of the available data so far and it can be seen that even after applying SEA, the classification accuracies are very similar as compared to training without data sampling thus proving that SEA is not only useful for incremental training but also for static training of deep learning models.

Apart from the standard deviations, in Fig. 4, we also show the overall reduction in both the sample transmission rate to the server and the training time of $ANN_{server}$ after applying SEA. We also show the average change in the classification accuracies after applying SEA. From Fig. 4, it can be seen that when evaluating SEA on several datasets, the sample transmission reduction rate to the server and training time reduction rate on the server is greater than 13% and 11% respectively in all cases. The maximum reduction in sample transmission rate to the server and training time reduction rate on the server we achieve is 15.94% and 23.67%, respectively. The maximum and minimum change in the average difference between the classification accuracies is only 1.81% and 0.56%, respectively. This shows that SEA can largely maintain incremental learning performance while greatly reducing the samples being transmitted to the server and the training time on the server.

## DISCUSSION

From the results obtained in Tables 2–5, SEA has the highest classification accuracies at the majority of the incremental training rounds across all datasets. WRST (*Ozcan & Basturk, 2020*), MT (*Alippi, Boracchi & Roveri, 2017*), and KWT (*Höpken et al., 2020*) lead to an extremely poor classification accuracy performance when evaluated on CUB-200 (*Welinder et al., 2010*) which indicates that such methods are inducing a lot of catastrophic forgetting in incremental learning as compared to SEA. Furthermore, SEA requires no hyperparameters for performing data sampling whereas WRST (*Ozcan & Basturk, 2020*), MT (*Alippi, Boracchi & Roveri, 2017*), and KWT (*Höpken et al., 2020*) all require the $p$-value to be set for performing data sampling.

The biggest advantage offered by SEA is in the reduction of the training samples followed by the reduction in the training time of the classifier on the server. In the context of fog computing, this leads to a big reduction in the communication cost between the fog device and the server along with an accelerated training time on the server. All of this is achieved while maintaining the model performance. In terms of the classification accuracies, what matters is the performance of the model after incrementally learning all the classes, in which case even after applying SEA, all the classification accuracies after the final incremental training round are less than 3% compared to the classification accuracies obtained where no data sampling is used. In our methodology, we discard $\rho_c$ samples whereby $\rho_c$ samples in each class have entropies whose distance from the median of the entropy distribution is smaller than the standard deviation of the entropy distribution of that class. We claim that due to this phenomenon, $\rho_c$ samples are not needed for training due to their extremely low variance with respect to the entropy distribution of the class as such samples will not aid the training of the neural network on the server. The results obtained in Table 6 support the theory of our data sampling algorithm (SEA).

Applying SEA on small-scale datasets is challenging as there is not enough room for data sampling and even if data sampling does take place, the number of samples to be discarded should be chosen very carefully as a huge impact can be made on incremental learning in such cases. For e.g., discarding one image from a total of hundred images results in only a 1% data discard rate but if one image is discarded from a total of ten images then this results in a 10% data discard rate. Despite this, our proposed SEA is able to retain incremental learning performance for small datasets such as CUB-200 (*Welinder et al., 2010*).

The reason why SEA works so well in terms of retaining incremental learning performance is simply because of the unpredictability of the novel classes themselves. When neural networks face novel classes and a cross-entropy loss is calculated for each image, the neural network can never guarantee that the loss values of all samples will decrease after training. On the contrary, it is possible that after training, the loss values of some samples might increase, this is especially true in the case of class incremental learning as seen in Fig. 5.

Figure 5A shows the initial loss values of samples belonging to classes with labels: 0-9 inclusive of CIFAR-100 (*Krizhevsky, 2009*), Fig. 5B shows the initial loss values of samples belonging to classes with labels: 50–59 of CIFAR-100 (*Krizhevsky, 2009*), and Fig. 5C shows

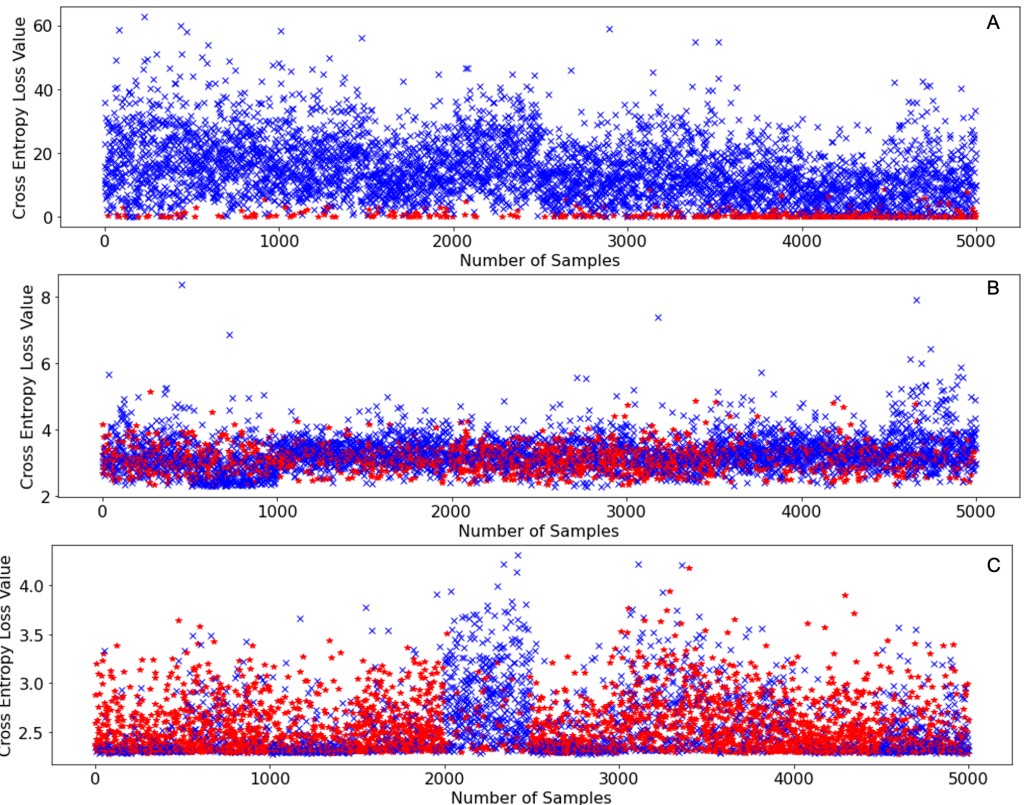

**Figure 5   Initial cross entropy values of samples from CIFAR-100 (*Krizhevsky, 2009*) on SqueezeNet (*Iandola et al., 2016*) when learning 10 classes at a time using the same learning settings as shown in Table 1.** (A) Initial cross entropy loss values of all samples with labels between 0–9 inclusive. (B) Initial cross entropy loss values of all samples with labels between 50–59. (C) Initial cross entropy loss values of all samples with labels between 90–99.

the initial loss values of samples belonging to the classes with labels: 90–99 inclusive of CIFAR-100 (*Krizhevsky, 2009*). The points in red denote the loss values of samples that increased after an incremental training round, the points in blue denote the loss values of samples that decreased after an incremental training round. From Fig. 5A it can be deduced that samples that have very low loss values are the ones that exhibit higher loss value after training. However, as incremental learning progresses, this no longer is the case as shown in Fig. 5B and Fig. 5C. As incremental learning progresses, the pattern at which the samples in every incremental training round that exhibit higher loss values (red marks) after training become increasingly randomized.

Since the occurrence of entropy values that increase after training (red points) becomes increasingly randomized as incremental learning progresses, we must discard a certain number of samples in a randomized manner as this increases the probability of discarding a sample that might not turn out useful for incremental learning. Therefore, SEA is the most consistent performing data sampling algorithm as we saw in the experiments that we conducted.

# CONCLUSIONS

In this paper, we present a novel data sampling technique called Sample Elimination Algorithm (SEA). We compare our algorithm with three non-parametric statistical tests namely: the Wilcoxon Rank-Sum Test (WRST) (*Ozcan & Basturk, 2020*), the Median Test (MT) (*Alippi, Boracchi & Roveri, 2017*), and the Kruskal-Wallis Test (KWT) (*Höpken et al., 2020*). Though SEA does not always lead to the least number of samples being transmitted to the server and the fastest training time on the server, SEA does lead to the highest classification accuracies at the majority of the incremental training rounds regardless of the dataset which is what set as the original target in this paper. We show that SEA is a more generalized data sampling algorithm as compared to the other non-parametric statistical tests for data sampling.

From the results obtained, we can conclude that our proposed SEA can adapt to input datasets of various sizes, different training hyperparameters, a different number of classes being learned incrementally, and still maintain incremental learning performance despite using fewer training images as compared to the total available samples. This in turn reduces the transmission costs to the server and the training time on the server as well. This is the case when working with the convolutional layers of CNNs for feature extraction of images and using Artificial Neural Networks (ANN) as classifiers for training.

## Limitations and Future Work

Due to the scope of this work, we use ANNs as classifiers, however, future work should investigate using classifiers other than ANNs. When discussing Fig. 1, we mentioned accumulating all the samples that belong to an incremental training round, however, storage is an important issue especially in the case of a large number of samples per class or a large number of classes in an incremental training round. Since we do not consider this concept in this work, for future work, it is important to perform data sampling as soon as a significant amount of memory is starting to get consumed on fog devices and knowing exactly when to start the incremental training process.

In this work, we consider one fog device and one server. Therefore, one potential future research direction is how to perform incremental learning in a distributed computing scenario i.e., multiple fog devices receiving new data classes.

We think that one of the most important advancements that can be made is the development of an unsupervised concept drift detection algorithm whereby such an algorithm should be able to know exactly when a new class has been encountered. Being able to do so greatly reduces the amount of catastrophic forgetting in incremental learning.

### Funding

This work was supported by the University of Nottingham Malaysia Campus and the Fundamental Research Grant Scheme (FRGS) by the Ministry of Higher Education, Malaysia (FRGS/1/2018/ICT02/UNIM/02/4). The funders had no role in study design, data collection and analysis, decision to publish, or preparation of the manuscript.

### Grant Disclosures

The following grant information was disclosed by the authors:
Ministry of Higher Education, Malaysia: FRGS/1/2018/ICT02/UNIM/02/4.

### Competing Interests

The authors declare there are no competing interests.

### Author Contributions

- Swaraj Dube conceived and designed the experiments, performed the experiments, analyzed the data, performed the computation work, prepared figures and/or tables, authored or reviewed drafts of the paper, and approved the final draft.
- Yee Wan Wong and Hermawan Nugroho conceived and designed the experiments, analyzed the data, prepared figures and/or tables, authored or reviewed drafts of the paper, and approved the final draft.

### Data Availability

The data and codes are available in the Supplemental Files and at figshare: Dube, Swaraj; Wong, Yee Wan; Nugroho, Hermawan (2021): Codes_and_data.zip. figshare. Dataset. https://doi.org/10.6084/m9.figshare.14872647.v1.

### Supplemental Information

Supplemental information for this article can be found online at http://dx.doi.org/10.7717/peerj-cs.633#supplemental-information.

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
