# Peer review of "Dynamic sampling of images from various categories for classification based incremental deep learning in fog computing"

_PeerJ Computer Science, doi:10.7717/peerj-cs.633_

## Round 0.1 · original submission · Major Revisions

Three reports have been received for the paper. The reviewers have identified some drawbacks such as flaws of design and insufficient implementation. Please provide point by point detailed responses. Please note you should not cite references recommended by the reviewers if they are not appropriate. Not using such references will not affect our editorial decision.

Reviewer 1 ·

Basic reporting

I am not sure what kind of habit you have while you were doing your literature review.
e.g.
This has led to state-of-the-art developments (Choi et al., 2019; Gepperth & Karaoguz, 2016; Hayes et al., 2019; Kemker & Kanan, 2018; Lesort et al., 2020; Nallaperuma et al., 2019; Parisi et al., 2018; Rebuffi et al., 2017; Rusu et al., 2016; Wu et al., 2018).
e.g.
A lot of research has been done in this area that discards training data 128 during training (Alain et al., 2015; Gopal, 2016; Katharopoulos & Fleuret, 2018; Loshchilov & 129 Hutter, 2015; Needell et al., 2016)
e.g.
achieving state-of-the-art classification accuracies (Castro et al., 2018; Kemker & Kanan, 2018; Rebuffi et al., 2017; Rusu et al., 2016; Wu et al., 2019)

a general statement + a list of people's names and years... it does not give the reader anything but just shows off that you seem to read some papers... yet who knows whether you actually read those...

a bad habit... if you actually went thru those, please summarise and present them!

Experimental design

1) there are figures of general knowledge on fog computing and fog device to the central server. but what is your design? figure 2 is too general, where CNN happens? where "transfer learning" happens?
2) i would like to know how you implement transfer learning.
3) the novelty might be the Sample Elimination Algorithm, but why you need it? what is/are its purpose? why do you want to discard samples in a class?

Validity of the findings

if it is a research of computer vision/image processing. at least show some qualitative result of your study.

Reviewer 2 ·

Basic reporting

In this article the author present few introduction in abstract which doesn't provide a link to proposed methodology. If possible few points may be added in abstract.

Literature was well organised.

Content of the paper is good.

Experimental design

Need more data samplings

Result present is not sufficient.

Axes parameters are not clear in Graphical representation.

Validity of the findings

Novel approach can be seen in this paper.

Additional comments

Provide the list of contributions in the Introduction

Most of the formulae in this article are considered from the published articles, recommended to cite them.

There are several grammatical errors and typos. It should be rectified.

Tables should be presented in understandable way. It look complicated.

Reviewer 3 ·

Basic reporting

Authors should explain the scenarios (e.g. live face recognition, healthcare etc.) where images are sent to fog devices and then forwarded to servers to provide readers a clear understanding of why fog computing is being employed here.


Some grammatical mistakes.

Experimental design

In fog computing many fog devices are connected to a server. The author should explain how incremental learning is performed by the server on receiving samples from the various connected fog devices.

The proposed sampling method is compared with 4 old statistical sampling methods. Authors may provide comparison of their method with some non-statistical, heuristic based selective sampling methods to ensure its effectiveness.

Validity of the findings

No comment

Additional comments

The technical details are well presented in the paper.

---

## Round 0.2 · accepted · Accept

The revised paper is now balanced. All reviewers comments have been addressed. I recommend it for publication.